# LiftPool: Bidirectional ConvNet Pooling

**Jiaojiao Zhao & Cees G. M. Snoek**
Video & Image Sense Lab
University of Amsterdam
{jzhao3,cgmsnoek}@uva.nl

## Abstract

Pooling is a critical operation in convolutional neural networks for increasing receptive fields and improving robustness to input variations. Most existing pooling operations downsample the feature maps, which is a lossy process. Moreover, they are not invertible: upsampling a downscaled feature map can not recover the lost information in the downsampling. By adopting the philosophy of the classical Lifting Scheme from signal processing, we propose LiftPool for bidirectional pooling layers, including LiftDownPool and LiftUpPool. LiftDownPool decomposes a feature map into various downsized sub-bands, each of which contains information with different frequencies. As the pooling function in LiftDownPool is perfectly invertible, by performing LiftDownPool backward, a corresponding up-pooling layer LiftUpPool is able to generate a refined upsampled feature map using the detail sub-bands, which is useful for image-to-image translation challenges. Experiments show the proposed methods achieve better results on image classification and semantic segmentation, using various backbones. Moreover, LiftDownPool offers better robustness to input corruptions and perturbations.

## 1 Introduction

Spatial pooling has been a critical ConvNet operation since its inception (Fukushima, 1979; LeCun et al., 1990; Krizhevsky et al., 2012; He et al., 2016; Chen et al., 2018). It is crucial that a pooling layer maintains the most important activations for the network's discriminability (Saeedan et al., 2018; Boureau et al., 2010). Several simple operations, such as average pooling or max pooling, have been explored for aggregating features in a local area. Springenberg et al. (2015) employ a convolutional layer with an increased stride to replace a pooling layer, which is equivalent to downsampling. While effective and efficient, simply using the average or maximum activation may ignore local structures. In addition, as these functions are not invertible, upsampling the downscaled feature maps can not recover the lost information. Different from existing pooling operations, we propose in this paper a bidirectional pooling called LiftPool, including LiftDownPool which preserves details when downsizing the feature maps, and LiftUpPool for generating finer upsampled feature maps.

LiftPool is inspired by the classical Lifting Scheme (Sweldens, 1998) from signal processing, which is commonly used for information compression (Pesquet-Popescu & Bottreau, 2001), reconstruction (Dogiwal et al., 2014), and denoising (Wu et al., 2004). The perfect invertibility of the Lifting Scheme stimulates some works on invertible networks (Dinh et al., 2017; Jacobsen et al., 2018; Atanov et al., 2019; Izmailov et al., 2020) . The Lifting Scheme decomposes an input signal into various sub-bands with downscaled size and this process is perfectly invertible. Applying the idea of Lifting Scheme, LiftDownPool factorizes an input feature map into several downsized spatial sub-bands with different correlation structures. As shown in Figure 1, for an image feature map, the $LL$ sub-band is an approximation removing several details. The $LH$, $HL$ and $HH$ represent details along horizontal, vertical and diagonal directions. LiftDownPool respects preserving any sub-band(s) as the pooled result. Moreover, due to the invertibility of the pooling function, LiftUpPool is introduced for upsampling feature maps. Upsampling a feature map is more challenging as seen for the MaxUpPool (Badrinarayanan et al., 2017), which generates an output with many 'holes' (shown in Figure 1). LiftUpPool utilizes the recorded details to recover a refined output by performing LiftDownPool backwards.

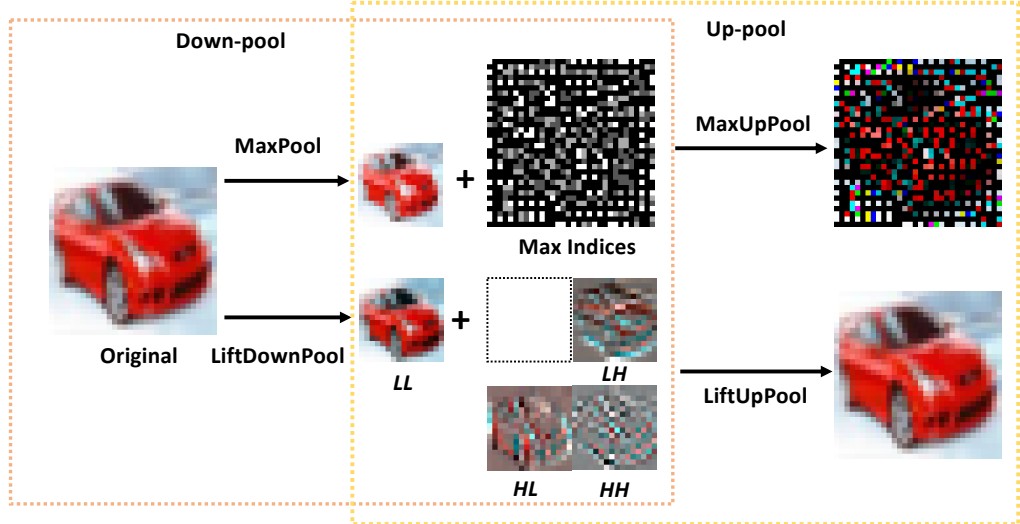

Figure 1: **Illustration of the proposed LiftDownPool and LiftUpPool** *vs.* MaxPool and MaxUp-Pool on an image from CIFAR-100. Where MaxPool takes the maximum activations from the input, LiftDownPool decomposes the input into four sub-bands: $LL$, $LH$, $HL$ and $HH$. $LL$ contains low-pass coefficients. It better reduces aliasing compared to MaxPool. $LH$, $HL$ and $HH$ represent details along horizontal, vertical and diagonal directions. For simplicity, we just upsample the down-pooled results for illustrating the up-pooling. MaxUpPool generates a sparse map with lost details. LiftUpPool produces a refined output from the recorded details by performing LiftDownPool backwards.

We analyze the proposed LiftPool from several viewpoints. LiftDownPool allows a flexible choice for any sub-band(s) as the pooled result. It outperforms baselines on image classification with various ConvNet backbones. LiftDownPool also presents better stability to corruptions and perturbations of inputs. By performing LiftDownPool backwards, LiftUpPool generates a refined up-sampling feature map for semantic segmentation.

## 2 METHODS

The down-pooling operator is formulated as minimizing the information loss caused by downsizing feature maps, as in image downscaling by Saeedan et al. (2018); Kobayashi (2019a). The Lifting Scheme (Sweldens, 1998) naturally matches the problem. The Lifting Scheme was originally designed to exploit the correlated structures present in signals to build a downsized approximation and several detail sub-bands in the spatial domain (Daubechies & Sweldens, 1998). The inverse transform is realizable and always provides a perfect reconstruction of the input. LiftPool is derived from the Lifting Scheme for bidirectional pooling layers.

### 2.1 LIFTDOWNPOOL

Taking a one-dimension (1D) signal as an example, LiftDownPool decomposes a given signal $\boldsymbol{x}=[x_1, x_2, x_3, ..., x_n], x_n \in \mathbb{R}$ into a *downscaled* approximation signal $\boldsymbol{s}$ and a difference signal $\boldsymbol{d}$ by,

$$\boldsymbol{s}, \boldsymbol{d} = F(\boldsymbol{x}). \tag{1}$$

where $F(\cdot)=f_{\text{update}} \circ f_{\text{predict}} \circ f_{\text{split}}(\cdot)$, consisting of three functions: split (downsample), predict and update. Here $\circ$ indicates the function composition operator. The LiftDownPool-1D is illustrated in Figure 2(a). Specifically,

**Split** $f_{\text{split}} : \boldsymbol{x} \mapsto (\boldsymbol{x}^e, \boldsymbol{x}^o)$. The given signal $\boldsymbol{x}$ is split into two disjoint sets $\boldsymbol{x}^e=[x_2, x_4, ..., x_{2k}]$ with even indices and $\boldsymbol{x}^o=[x_1, x_3, ..., x_{2k+1}]$ with odd indices. The two sets are typically closely correlated.

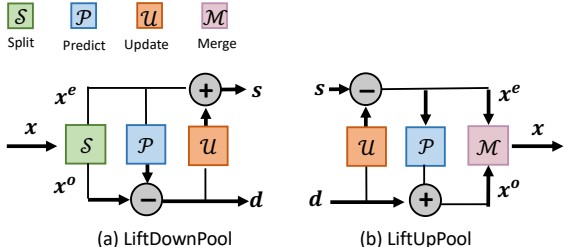

Figure 2: **LiftDownPool and Lift-UpPool implementations.** (a) LiftDownPool-1D. $\boldsymbol{x}$ is split into $\boldsymbol{x}^e$ and $\boldsymbol{x}^o$. The Predictor and Updater generate details $\boldsymbol{d}$ and an approximation $\boldsymbol{s}$. (b) LiftUpPool-1D. By running LiftDownPool backwards, $\boldsymbol{x}^e$ and $\boldsymbol{x}^o$ are generated from $\boldsymbol{s}$ and $\boldsymbol{d}$, and then merged into $\boldsymbol{x}$.

**Predict** $f_{\text{predict}} : (\boldsymbol{x}^e, \boldsymbol{x}^o) \mapsto \boldsymbol{d}$. Given one set *e.g.* $\boldsymbol{x}^e$, another set $\boldsymbol{x}^o$ is able to be predicted by a predictor $\mathcal{P}(\cdot)$. The predictor is not required to be precise, so the difference with the high-pass coefficients $\boldsymbol{d}$ is defined as:

$$\boldsymbol{d} = \boldsymbol{x}^o - \mathcal{P}(\boldsymbol{x}^e). \tag{2}$$

**Update** $f_{\text{update}} : (\boldsymbol{x}^e, \boldsymbol{d}) \mapsto \boldsymbol{s}$. Taking $\boldsymbol{x}^e$ as an approximation of $\boldsymbol{x}$ causes a serious aliasing because $\boldsymbol{x}^e$ is simply downsampled from $\boldsymbol{x}$. Particularly, the running average of $\boldsymbol{x}^e$ is not the same as that of $\boldsymbol{x}$. To correct it, a smoothed version $\boldsymbol{s}$ is generated by adding $\mathcal{U}(\boldsymbol{d})$ to $\boldsymbol{x}^e$:

$$\boldsymbol{s} = \boldsymbol{x}^e + \mathcal{U}(\boldsymbol{d}). \tag{3}$$

The update procedure is equivalent to applying a low-pass filter to $\boldsymbol{x}$. Thus, $\boldsymbol{s}$ with low-pass coefficients is taken as an approximation of the original signal.

The classific Lifting Scheme method applies pre-defined low-pass filters and high-pass filters to decompose an image into four sub-bands. However, pre-designing filters in $\mathcal{P}(\cdot)$ and $\mathcal{U}(\cdot)$ is difficult (Zheng et al., 2010). Earlier, Zheng et al. (2010) proposed to optimize these filters by a back-propagation network. All functions in LiftDownPool are differentiable. $\mathcal{P}(\cdot)$ and $\mathcal{U}(\cdot)$ are able to be simply implemented by convolution operators followed by non-linear activation functions (Glorot et al., 2011). Specifically, we design $\mathcal{P}(\cdot)$ and $\mathcal{U}(\cdot)$ as:

$$\mathcal{P}(\cdot) = \text{Tanh}() \circ \text{Conv}(\text{k=1,s=1,g=}G_2) \circ \text{ReLU}() \circ \text{Conv}(\text{k=}K\text{,s=1,g=}G_1), \tag{4}$$

$$\mathcal{U}(\cdot) = \text{Tanh}() \circ \text{Conv}(\text{k=1,s=1,g=}G_2) \circ \text{ReLU}() \circ \text{Conv}(\text{k=}K\text{,s=1,g=}G_1). \tag{5}$$

Here $K$ is the kernel size and $G_1$ and $G_2$ are the number of groups. We prefer to learn the filters in $\mathcal{P}(\cdot)$ and $\mathcal{U}(\cdot)$ with deep neural networks in an end-to-end fashion. To that end, two constraints need to be added to the final loss function. Recall, $\boldsymbol{s}$ is the downsized approximation of $\boldsymbol{x}$. As $\boldsymbol{s}$ is updated from $\boldsymbol{x}^e$ according to Eq 3, $\boldsymbol{s}$ is essentially close to $\boldsymbol{x}^e$. Thus, $\boldsymbol{s}$ is naturally required to be close to $\boldsymbol{x}^o$ as well. Therefore, one constraint term $c_u$ is for minimising the L2-norm distance between $\boldsymbol{s}$ and $\boldsymbol{x}^o$. With Eq 3,

$$\begin{aligned} c_u &= \|\boldsymbol{s} - \boldsymbol{x}^o\|_2 \\ &= \|\mathcal{U}(\boldsymbol{d}) + \boldsymbol{x}^e - \boldsymbol{x}^o\|_2. \end{aligned} \tag{6}$$

The other constraint term $c_p$ is for minimising the detail $\boldsymbol{d}$, with Eq 2,

$$c_p = \|\boldsymbol{x}^o - \mathcal{P}(\boldsymbol{x}^e)\|_2. \tag{7}$$

The total loss is:

$$\mathcal{L} = \mathcal{L}_{\text{task}} + \lambda_u c_u + \lambda_p c_p, \tag{8}$$

where $\mathcal{L}_{\text{task}}$ is the loss for a specific task, like a classification or semantic segmentation loss. We set $\lambda_u$=0.01 and $\lambda_p$=0.1. Our experiments show the two terms bring good regularization to the model.

LiftDownPool-2D is easily decomposed into several 1D LiftDownPool operators. Following the standard Lifting Scheme, we first perform a LiftDownPool-1D along the horizontal direction to obtain an approximation part $\boldsymbol{s}$ (low frequency in the horizontal direction) and a difference part $\boldsymbol{d}$ (high frequency in the horizontal direction). Then, for each of the two parts, we apply the same LiftDownPool-1D along the vertical direction. By doing so, $\boldsymbol{s}$ is further decomposed into $LL$ (low frequency in vertical and horizontal directions) and $LH$ (low frequency in the vertical direction and high frequency in the horizontal direction). $\boldsymbol{d}$ is further decomposed into $HL$ (high frequency in the vertical direction and low frequency in the horizontal direction) and $HH$ (high frequency in vertical

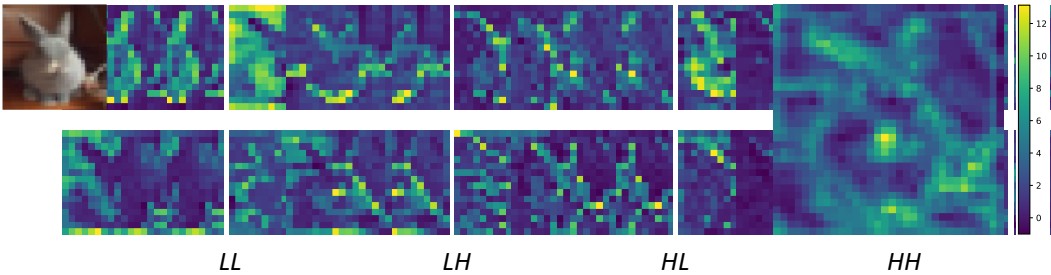

$LL$        $LH$        $HL$        $HH$

Figure 3: **LiftDownPool visualization.** Selected feature maps of an image in CIFAR-100, from the first LiftDownPool layer in VGG13. $LL$ represents smoothed feature maps with less details. $LH$, $HL$ and $HH$ represent detailed features along horizontal, vertical and diagonal directions. Each sub-bands contains different correlation structures.

and horizontal directions). We can flexibly choose sub-band(s) for down-pooling and keep the other sub-band(s) for reversing the operation. Naturally, LiftDownPool-1D can be generalized further for any $n$-dimensional signal. In Figure 3, we show several feature maps from the first LiftDownPool layer based on VGG13. $LL$ has smoothed features with less details. $LH$, $HL$ and $HH$ capture the details along horizontal, vertical and diagonal directions.

**Discussion** MaxPool is usually formulated as first performing Max and then downsampling: $\text{MaxPool}_{k,s} = \text{downsample}_s \circ \text{Max}_k$ (Zhang, 2019). By contrast, LiftDownPool is: $\text{LiftDownPool}_{k,s} = \text{update}_k \circ \text{predict}_k \circ \text{downsample}_s$. First downsampling and then performing two lifting steps (prediction and updating) helps anti-aliasing. A simple analysis is provided in the Appendix. As shown in Figure 1, LiftDownPool keeps more structured information and better reduces aliasing then MaxPool.

## 2.2 LIFTUPPOOL

LiftPool inherits the invertibility of the Lifting Scheme. Taking the 1D signal as an example, LiftUpPool generates an upsampled signal $x$ from $s, d$ by:

$$x = \mathcal{G}(s, d). \tag{9}$$

where $\mathcal{G}(\cdot) = f_{\text{merge}} \circ f_{\text{predict}} \circ f_{\text{update}}(\cdot)$, including the functions: update, predict and merge. Specifically, $s, d \mapsto x^e, d \mapsto x^e, x^o \mapsto x$ are realized by:

$$x^e = s - \mathcal{U}(d), \tag{10}$$

$$x^o = d + \mathcal{P}(x^e), \tag{11}$$

$$x = f_{\text{merge}}(x^e, x^o). \tag{12}$$

We simply get the even part $x^e$ and odd part $x^o$ from $s$ and $d$, and then merge $x^e$ and $x^o$ into $x$. In this way, we generate upsampled feature maps with rich information.

**Discussion** Up-pooling has been used in image-to-image translation tasks such as semantic segmentation (Chen et al., 2017), super-resolution (Shi et al., 2016), and image colorization (Zhao et al., 2020). It is generally used in encoder-decoder networks such as SegNet (Badrinarayanan et al., 2017) and UNet (Ronneberger et al., 2015). However, most existing pooling functions are not invertible. Taking MaxPool as the baseline, it is required to record the maximum indices during max pooling. For simplicity, we use the down-pooled results as inputs to the up-pooling in Figure 1. When performing MaxUpPool, the values of the input feature maps are directly filled on the corresponding maximum indices of the output and other indices will be given zeros. By doing so, the output looks sparse and loses much of the structured information, which is harmful for generating good-resolution outputs. LiftUpPool performing an inverse transformation of LiftDownPool, is able to produce finer outputs by using the multiphase sub-bands.

## 3 RELATED WORK

Taking the average over a feature map region was the pooling mechanism of choice in the Neocognitron (Fukushima, 1979; 1980) and LeNet (LeCun et al., 1990). Average pooling is equivalent to blurred-downsampling. Max pooling later proved even more effective (Scherer et al., 2010) and became popular in deep ConvNets. Yet, averaging activations or picking the maximum activations causes loss of details. Zeiler & Fergus (2013) and Zhai et al. (2017) introduced a stochastic process to pooling and downsampling, respectively, for a better regularization. Lee et al. (2016) mixed AveragePool and MaxPool by a gated mask to adapt to complex and variable input patterns. Saeedan et al. (2018) introduced detail-preserving pooling (DPP) for maintaining structured details. By contrast, Zhang (2019) proposed a BlurPool by applying a low-pass filter, which removes details. Interestingly, both methods improved image classification, indicating that (empirically) determining the best pooling strategy is beneficial (Saeedan et al., 2018). Williams & Li (2018) introduced the wavelet transform into pooling for reducing jagged edges and other artifacts. Rippel et al. (2015) suggested pooling in the frequency domain, which enabled flexibility in the choice of the pooling output dimensionality. Pooling based on a probabilistic model was proposed in (Kobayashi, 2019a) and (Kobayashi, 2019b). Kobayashi (2019a) first used a Gaussian distribution to model the local activations and then aggregates the activations into the two statistics of mean and standard deviation. Kobayashi (2019b) estimated parameters from global statistics in the input feature map, to flexibly represent various pooling types. Our proposed LiftDownPool decomposes the input feature map into a downsized approximation and several details. It is flexible to choose any sub-band(s) as pooled result.

While existing pooling functions are not invertible, our proposed LiftPool is able to perform both down-pooling and up-pooling. Previously, MaxUpPool (Badrinarayanan et al., 2017) was introduced for semantic segmentation. As the max pooling function is not invertible, the lost details can not be recovered during up-pooling. Hence, the output suffers from aliasing. Although adding a BlurPool to MaxUpPool may help to reduce the aliasing (Zhang, 2019), several details are still lost. LiftUpPool, performing the LiftDownPool functions backwards, is capable of producing a refined high-resolution output with the help of the details sub-bands.

Earlier, Zheng et al. (2010) introduce back-propagation for the Lifting Scheme to perform nonlinear wavelet decomposition. They propose an update-first Lifting Scheme and use back-propagation to replace the Updater and Predictor in the Lifting Scheme. In this way, they realize a back-propagation neural network in lifting steps for signal processing. There is no pooling layer used. We develop down-pooling and up-pooling layers by leveraging the idea of the Lifting Scheme for image processing. We utilize convolution layers and ReLU layers to implement the Updater and Predictor, which are optimized end-to-end with the deep neural network. Our pooling layers are easily plugged into various backbones. Recently, Rodriguez et al. (2020) introduce the Lifting Scheme for multiresolution analysis in a network. Specifically, they develop an adaptive wavelet network by stacking several convolution layers and Lifting Scheme layers. They focus on an interpretable network by integrating multiresolution analysis, rather than pooling. Our paper aims at developing a pooling layer by utilizing the lifting steps. We develop a down-pooling that constructs various sub-bands with different information, and an up-pooling which generates refined upsampled feature maps.

## 4 EXPERIMENTS

### 4.1 CONVNET TESTBEDS

**Image Classification** We first verify the proposed LiftDownPool for image classification on **CIFAR-100** (Krizhevsky & Hinton, 2009) with $32 \times 32$ low-resolution images. CIFAR-100 has 100 classes with 600 images each. There are 500 training images and 100 testing images per class. A VGG13 (Simonyan & Zisserman, 2015) network is trained on this dataset. For experiments conducted on CIFAR-100, we repeat each experiment three times with different initial random seeds during training and report the averaged error rate with the standard deviation. We also report results on **ImageNet** (Deng et al., 2009) with 1.2M training and 5000 validation images for 1000 classes. We plug the LiftDownPool into several popular ConvNet backbones to verify its generalizability for image classification. We replace the local pooling layers by LiftDownPool in all the networks. Error rate is utilized as the evaluation metric. All training settings are provided in the Appendix.

Table 1: **Flexibility.** Top-1 image classification error rate with varying sub-bands on CIFAR-100. Mixing low-pass and high-pass obtains the best result. Adding $c_u$ and $c_p$ helps improve the result.

|  | *Top-1* |
|---|---|
| $LL$ | $25.64 \pm 0.04$ |
| $LH$ | $25.71 \pm 0.04$ |
| $HL$ | $24.88 \pm 0.05$ |
| $HH$ | $25.18 \pm 0.08$ |
| $LL+LH+HL+HH$ (w/o $c_u$ and $c_p$) | $26.43 \pm 0.07$ |
| $LL+LH+HL+HH$ (w/ $c_u$ and $c_p$) | $\textbf{24.35} \pm 0.11$ |

| Kernel | *Top-1* |
|---|---|
| 2 | $25.53 \pm 0.13$ |
| 3 | $25.06 \pm 0.22$ |
| 4 | $24.89 \pm 0.07$ |
| 5 | $\textbf{24.35} \pm 0.11$ |
| 7 | $24.40 \pm 0.08$ |

|  | *Top-1* |
|---|---|
| Skip | $27.09 \pm 0.11$ |
| MaxPool | $25.71 \pm 0.13$ |
| AveragePool | $25.87 \pm 0.03$ |
| **LiftDownPool** | $\textbf{24.35} \pm 0.11$ |

Table 2: **Effectiveness.** Top-1 image classification error rate with varying kernel size on CIFAR-100. Kernel size 5 achieves better result.

Table 3: **Effectiveness.** Top-1 image classification error rate with various pooling methods on CIFAR-100. LiftDownPool outperforms baselines.

**Semantic Segmentation**   We also test the LiftDownPool and LiftUpPool for semantic segmentation on ***PASCAL-VOC12*** (Everingham et al., 2010), which contains 20 foreground object classes and one background class. An augmented version with 10582 training images and 1449 validation images is used. We consider SegNet (Badrinarayanan et al., 2017) with VGG13 and DeeplabV3Plus (Chen et al., 2018) with ResNet50 as ConvNets for semantic segmentation. The performance is measured in terms of pixel mean-intersection-over-union (*mIoU*) across the 21 classes. Code is available at https://github.com/jiaozizhao/LiftPool/.

## 4.2   ABLATION STUDY

**Flexibility**   We first test VGG13 on CIFAR-100. Different from previous pooling methods, LiftDownPool generates four sub-bands, each of which contains a different type of information. LiftDownPool allows to flexibly choose which sub-band(s) to keep as the final pooled results. In Table 1, we show the *Top-1* error rate for the classification based on different sub-bands. Interestingly, it is observed that vertical details contribute more for image classification. Low-pass coefficients and high-pass coefficients along horizontal direction get similar error rate. Whether the two spatial dimensions should be treated equally we leave for our future work. To realize a less lossy pooling, we combine all the sub-bands by summing them up with almost no additional compute cost. Such a pooling significantly improves the results. In addition, the constrains $c_u$ and $c_p$ help to decrease the error rate. Moreover, seen from Table 1 and Table 3, we further conclude LiftDownPool outperforms other baselines even based on any single sub-band. We believe the learned LiftDownPool provides an effective regularization to the model.

**Effectiveness**   Table 2 ablates the performance when varying kernel sizes for the filters in $\mathcal{P}(\cdot)$ and $\mathcal{U}(\cdot)$. A larger kernel size, covering more local information, performs slightly better. When kernel size equals 7, it brings more computations but no performance gain. Unless specified otherwise, we use for all experiments from now on a kernel size of 5 and we sum up all the sub-bands. We also compare our LiftDownPool with the commonly-used MaxPool, AveragePool, as well as the convolutional layer with stride 2 (Springenberg et al., 2015), which is called Skip by Kobayashi (2019a). Seen from Table 3, LiftDownPool outperforms other pooling methods on CIFAR-100.

**Generalizability**   We apply LiftDownPool to several backbones including ResNet18, ResNet50 (He et al., 2016) and MobileNet-V2 (Sandler et al., 2018) on ImageNet. In Table 4, LiftDownPool has 2% lower *Top-1* error rate than MaxPool and AveragePool. While combining MaxPool and AveragePool in a Gated (Lee et al., 2016) or Mixed (Lee et al., 2016) fashion, still has a 1% gap with LiftDownPool. Gauss (Kobayashi, 2019a) and GFGP (Kobayashi, 2019b) are comparable to LiftDownPool with ResNet50, but not with lighter networks. Compared

| | ResNet18 | | ResNet50 | | MobileNet-V2 | |
|---|---|---|---|---|---|---|
| | *Top-1* | *Top-5* | *Top-1* | *Top-5* | *Top-1* | *Top-5* |
| Skip (Springenberg et al., 2015) | 30.22 | 10.23 | 24.31 | 7.34 | 28.66 | 9.70 |
| MaxPool | 28.60 | 9.77 | 24.26 | 7.22 | 28.65 | 9.82 |
| AveragePool | 28.03 | 9.55 | 24.40 | 7.35 | 28.32 | 9.72 |
| $S^3$Pool (Zhai et al., 2017) | 33.91 | 13.09 | 27.98 | 9.34 | 40.56 | 17.91 |
| WaveletPool (Williams & Li, 2018) | 30.33 | 10.82 | 24.43 | 7.36 | 29.27 | 10.26 |
| BlurPool$^\star$ (Zhang, 2019) | 29.88 | 10.58 | 24.60 | 7.73 | 30.58 | 11.26 |
| DPP$^\star$ (Saeedan et al., 2018) | 29.12 | 10.21 | 24.62 | 7.49 | 29.85 | 10.53 |
| SpectralPool (Rippel et al., 2015) | 28.69 | 9.87 | 24.81 | 7.57 | 33.38 | 12.56 |
| GatedPool (Lee et al., 2016) | 27.78 | 9.44 | 23.79 | 7.06 | 28.94 | 9.90 |
| MixedPool (Lee et al., 2016) | 27.76 | 9.50 | 24.08 | 7.32 | 29.00 | 9.97 |
| GFGP$^\star$ (Kobayashi, 2019b) | 26.88 | 8.66 | 22.76 | 6.34 | 28.42 | 9.59 |
| GaussPool$^\star$ (Kobayashi, 2019a) | 26.58 | 8.86 | 22.95 | 6.30 | 27.13 | 8.92 |
| **LiftDownPool** | **25.80** | **8.14** | **22.36** | **6.11** | **26.09** | **8.22** |

Table 4: **Generalizability** of LiftDownPool on ImageNet. LiftDownPool outperforms alternative pooling methods, no matter what ConvNet backbone is used. $\star$ means the numbers are based on running the code provided by authors. Others are based on our re-implementation.

| | Normalized | | Unnormalized | |
|---|---|---|---|---|
| | ImageNet-C | ImageNet-P | ImageNet-C | ImageNet-P |
| | *mCE* | *mPR* | *mCE* | *mPR* |
| Skip | 72.71 | 61.75 | 57.05 | 7.56 |
| MaxPool | 73.09 | 62.64 | 57.40 | 7.57 |
| AveragePool | 72.09 | 56.23 | 56.56 | 6.90 |
| BlurPool (Zhang, 2019) | 72.14 | 56.54 | 56.58 | 6.90 |
| DPP (Saeedan et al., 2018) | 72.12 | 62.30 | 56.67 | 7.62 |
| GatedPool (Lee et al., 2016) | 72.58 | 58.05 | 57.00 | 7.23 |
| GaussPool (Kobayashi, 2019a) | 69.27 | 54.83 | 54.40 | 6.76 |
| **LiftDownPool** | **68.45** | **52.91** | **53.80** | **6.55** |

Table 5: **Out-of-distribution robustness** of LiftDownPool on ImageNet-C and ImageNet-P. Lift-DownPool is more robust to corruptions and perturbations compared to baselines.

to Spectral pooling (Rippel et al., 2015) and Wavelet pooling (Williams & Li, 2018), which are applied in the frequency or space-frequency domain, LiftDownPool offers an advantage by learning correlated structures and details in the spatial domain. Compared to DPP (Saeedan et al., 2018), which preserves details, and BlurPool (Zhang, 2019), smoothing feature maps by a low-pass filter, our LiftDownPool retains all sub-bands which proves to be more powerful for image classification. Stochastic approaches like $S^3$Pool obtain poor results on the large-scale dataset because randomness in pooling hampers network training, as earlier observed by Kobayashi (2019b). To conclude, LiftDownPool performs better no matter what backbone is used.

**Parameter Efficiency.** For all pooling layers in one network, we use the same kernel size in LiftPool. For the trainable parameters, recall $\mathcal{P}$ or $\mathcal{U}$ has a 1D convolution, so each has $C/G_1 \times C \times K + G_2$ parameters. $C$ is the number of the input channels and $G_2$ equals the number of internal channels. A 2D LiftDownPool shares these parameters three times without extra parameters. We compare our LiftDownPool (with 25.58 M) to two other parameterized pooling methods using ResNet50 on ImageNet: GFGP (31.08 M) and GaussPool (33.85 M). We achieve a lower error rate compared to GFGP and GaussPool with less parameters. Our performance boost is due to the LiftPool scheme, not the added capacity.

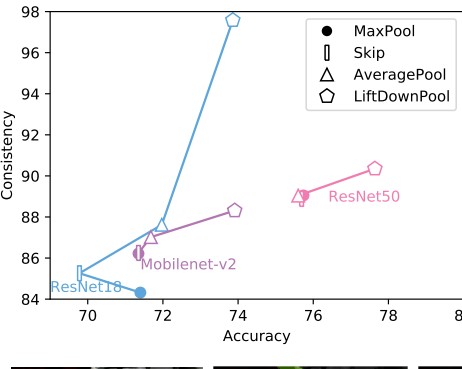

Figure 4: **Shift Robustness** comparisons between various pooling methods including MaxPool, Skip, AveragePool and the proposed LiftDownPool. LiftDownPool improves classification consistency and meanwhile boosts the accuracy, independent of the backbone used.

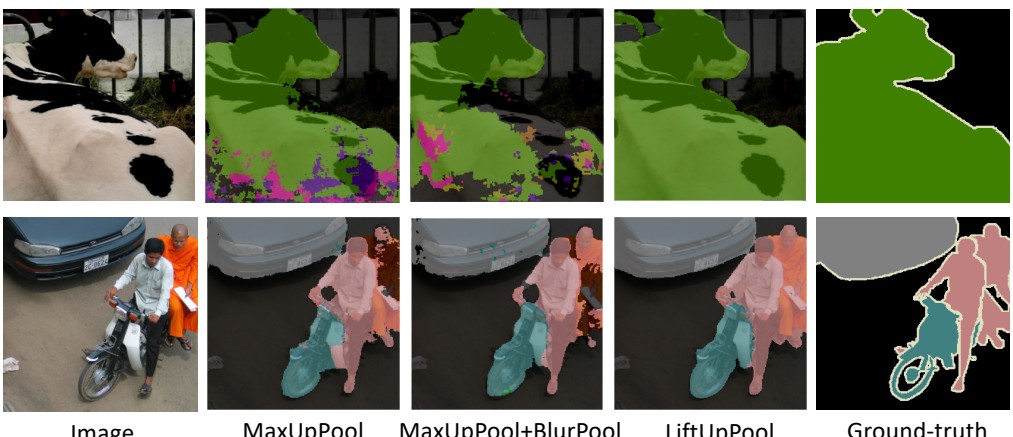

| Image | MaxUpPool | MaxUpPool+BlurPool | LiftUpPool | Ground-truth |

Figure 5: **LiftUpPool for Semantic Segmentation.** Visualization of semantic segmentation maps on PASCAL-VOC12 based on SegNet with varying up-pooling methods. LiftUpPool presents more completed, precise segmentation maps with smooth edges.

### 4.3    STABILITY ANALYSIS

**Out-of-distribution Robustness**    A good down-pooling method is expected to be stable to perturbations and noise. Following Zhang (2019), we test the robustness of LiftDownPool to corruptions on *ImageNet-C* and stability to perturbations on *ImageNet-P* using ResNet50. Both datasets come from Hendrycks & Dietterich (2019). We report the mean Corruption Error (*mCE*) and mean Flip Rate (*mFR*) for the two tasks, with both unnormalized raw values and normalized values by AlexNet's *mCE* and *mFR*, following Hendrycks & Dietterich (2019).

From Table 5, LiftDownPool effectively reduces raw *mCE* compared to the baselines. We show *CE* for each corruption type for further analysis in Figure 9 in the Appendix. LiftDownPool enables robustness to both "high-frequency" corruptions, such as noise or spatter, and "low-frequency" corruptions, like blur and jpeg compression. We believe LiftDownPool benefits from the mechanism that all sub-bands are used. A similar conclusion is obtained for robustness to perturbations on ImageNet-P from Table 5 and Figure 9 in the Appendix. ImageNet-P contains short video clips of a single image with small perturbations added. Such perturbations are generated by several types of noise, blur, geometric changes, and simulated weather conditions. The metric *FR* measures how often the Top-1 classification changes in consecutive frames. It is designed for testing a model's stability under small deformations. Again, LiftDownPool achieves lower *FR* for most perturbations.

**Shift Robustness**    We then test the shift-invariance of our model. Following Zhang (2019), we use classification consistency to measure the shift-invariance. It represents how often the network outputs the same classification, given the same image with two different shifts. We test the models with varying backbones trained on ImageNet. In Figure 4, LiftDownPool boosts classification accuracy as well as consistency no matter which backbone is used. Besides, we have other interesting findings. The deeper ResNet50 network has more stable shift-invariance. Various pooling methods including MaxPool, Skip, AveragePool, do not make significant difference on consistency. How-

|  | mIoU |
|---|---|
| MaxUpPool | 62.7 |
| MaxUpPool + BlurPool | 64.0 |
| **LiftUpPool** | **68.9** |

Table 6: **LiftUpPool for Semantic Segmentation** on PASCAL-VOC12 based on SegNet with varying up-pooling methods.

|  | mIoU |
|---|---|
| Skip | 76.1 |
| MaxPool | 76.2 |
| AveragePool | 76.4 |
| Gauss (Kobayashi, 2019a) | 77.4 |
| **LiftDownPool** | **78.7** |

Table 7: **Semantic Segmentation with DeepLabV3Plus** on PASCAL-VOC12 with various pooling methods. Lift-DownPool performs best.

ever, a lighter ResNet18 network is influenced much by the pooling method. LiftDownPool brings more than 10% improvement on consistency using ResNet18. We leave for future work how the depth of the network affects the shift-invariance of the network itself.

## 4.4 RESULTS FOR SEMANTIC SEGMENTATION

**LiftUpPool for Semantic Segmentation**   LiftDownPool functions are invertible as described in Eq 10 and Eq 11. It naturally benefits a corresponding up-pooling operation, which is popularly used in Encoder-Decoder networks for image-to-image translation tasks. Usually, the Encoder downsizes feature maps layer by layer to generate a high-level embedding for understanding the image. Then the Decoder needs to translate the embedding with a tiny spatial size to a required map with the same spatial size as the original input image. Interpreting details is pivotal for producing high-resolution outputs. We replace all down-pooling and up-pooling layers with LiftDownPool and LiftUpPool in SegNet for semantic segmentation on PASCAL-VOC12. For LiftDownPool we only keep the $LL$ sub-band. For LiftUpPool, the detail-preserving sub-bands $LH$, $HL$ and $HH$ are used for generating upsampled feature maps. MaxUpPool is taken as the baseline. We also test MaxUpPool followed by a BlurPool (Zhang, 2019), which is expected to help anti-aliasing. Table 6 reveals LiftUpPool improves over the baselines with a considerable margin. As illustrated in Figure 1, MaxUpPool is unable to compensate for the lost details. Although BlurPool helps smoothing local areas, it can only provide a small improvement. As LiftUpPool is capable of refining the feature map by fusing it with details, it is beneficial for per-pixel prediction tasks like semantic segmentation. We show several examples for semantic segmentation in Figure 5. LiftUpPool is more precise on details and edges. We also show the feature maps per predicted class in the Appendix.

**Semantic Segmentation with DeepLabV3Plus**   As discussed, LiftDownPool helps to lift Conv-Nets on accuracy and stability for image classification. ImageNet-trained ConvNets often serve as the backbones for downstream tuning. It is expected to transfer the nature of LiftDownPool to other tasks. We still consider semantic segmentation as our example. We leverage the state-of-the-art DeeplabV3Plus-ResNet50 (Chen et al., 2018). The input image has size $512{\times}512$. The output feature map of the encoder is $32{\times}32$. The decoder upsamples the feature map to $128{\times}128$ and concatenates them with the low-level feature map for the final pixel-level classification. As before, all local pooling layers are replaced by LiftDownPool. We use the pre-trained weights for image classification to initialize the corresponding model. As shown in Table 7, LiftDownPool outperforms all the baselines with considerable gaps.

## 5 CONCLUSION

Applying classical signal processing theory to modern deep neural networks, we propose a novel pooling method: LiftPool. LiftPool is able to perform both down-pooling and up-pooling. Lift-DownPool improves both accuracy and robustness for image classification. LiftUpPool, generating refined upsampling feature maps, outperforms MaxUpPool by a considerable margin on semantic segmentation. Future work may focus on applying LiftPool to fine-grained image classification, super-resolution challenges or other tasks with high demands for detail preservation.

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

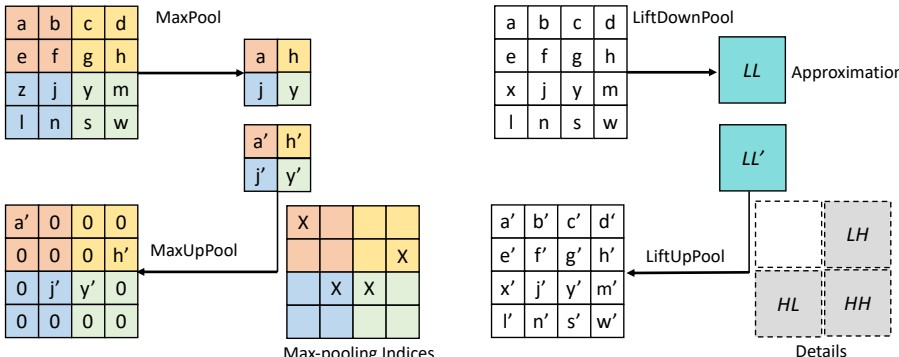

Figure 6: **Comparisons between MaxPool and LiftDownPool, MaxUpPool and LiftUpPool.**
MaxPool looses details. With the recorded maximum indices, MaxUpPool generates a very sparse
output. LiftDownPool decomposes the input into an approximation and several details sub-bands. It
realizes a pooling by summing up all sub-bands. LiftUpPool produces a refined output by perform-
ing LiftDownPool backwards.

# A    APPENDIX

We show additional analysis and results for robustness and semantic segmentation in this Appendix.

**LiftDownPool vs. MaxPool**    We provide a schematic diagram in Figure 6 to further illustrate the
difference between MaxPool and LiftDownPool, MaxUpPool and LiftUpPool. Taking kernel size 2,
stride 2 as an example, MaxPool selects the maximum activations in a local neighbourhood. Hence,
it looses 75% information. The lost details could be important for image recognition. LiftDownPool
decomposes a feature map into $LL$, $LH$, $HL$ and $HH$. $LL$ containing low-pass coefficients is an
approximation of the input. It is designed for capturing correlated structures of the input. Other
sub-bands contain detail coefficients along different directions. The pooling is implemented by
summing up all the sub-bands. The final pooled result containing both the approximation and details
is expected to be more effective for image classification.

**LiftUpPool vs. MaxUpPool**    The pooling function in MaxPool is not invertible. MaxPool records
the maximum indices for performing the corresponding MaxUpPool. MaxUpPool takes the acti-
vations at the corresponding positions for the recorded maximum indices on the output. For other
indices, there will be zeros. The final upsampled output has many 'holes'. By contrast, the pooling
functions in LiftDownPool are invertible. Leveraging the property by performing a LiftDownPool
backwards, LiftUpPool is able to generate a refined output from an input, including the recorded
details.

**Experiment Settings**    The VGG13 (Simonyan & Zisserman, 2015) network trained on CIFAR-
100 is optimized by SGD with a batch size of 100, weight decay of 0.0005, momentum of 0.9.
The learning rate starts from 0.1 and is reduced by multiplying 0.1 after 80 and 120 epochs for a
total of 160 epochs. We train ResNets for 100 epochs and MobileNet for 150 epochs on ImageNet,
following the standard training recipe from the public PyTorch (Paszke et al., 2017) repository.

**High-resolution Feature Maps Visualization**    By using ResNet50 with input size 224×224, we
extract the feature maps of an image from the first pooling layer. We show the high-resolution feature
maps in Figure 7. We only show the $LL$ sub-band from LiftDownPool. Compared to MaxPool,
LiftDownPool better maintains the local structure.

**Anti-aliasing**    LiftDownPool effectively reduces aliasing following the Lifting Scheme (Sweldens,
1998) compared to naive downsizing. Figure 8(b) provides a simple illustration of LiftDownPool.
The dashed line is an original signal $x$. According to Eq 1, the predictor $\mathcal{P}(\cdot)$ for the odd part $x_{2k+1}$
could easily take the average of its two even neighbors:

$$d_k = x_{2k+1} - (x_{2k} + x_{2k+2})/2 \qquad (13)$$

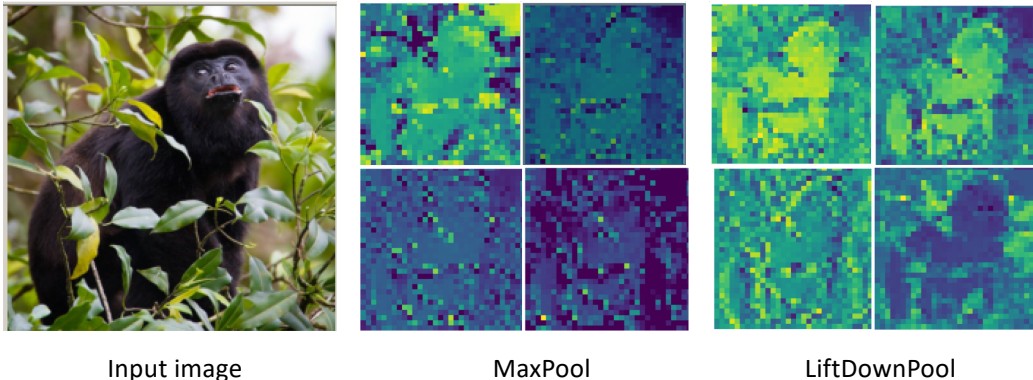

Figure 7: **High-resolution feature maps visualization.** LiftDownPool better maintains local structure.

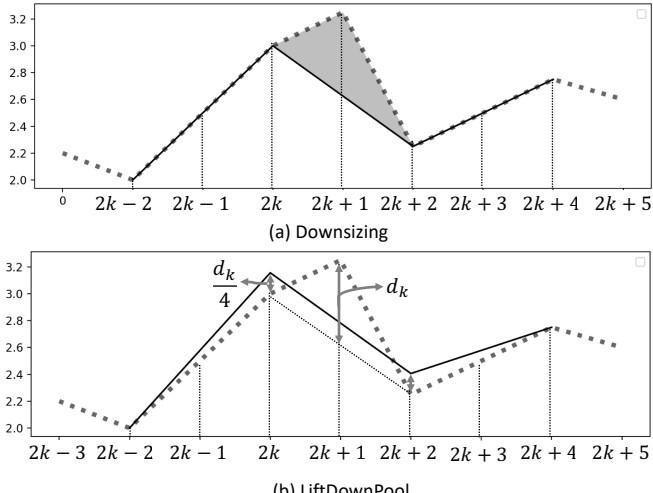

Figure 8: **Illustration how LiftDownPool reduces aliasing compared to downsizing** (Sweldens, 1998). Dashed line means original signal. (a) solid line is after downsizing. (b) solid line is after LiftDownPool. The solid and dashed lines cover the same area in (b).

Thus, if $x$ is linear in a local area, the detail $d_k$ is zero. The prediction step takes care of some of the spatial correlation. If an approximation $s$ of the original signal $x$ is simply taken from the even part $x^e$, it is really downsizing the signal shaped in the red line. There is serious aliasing. The running average of $x^e$ is not the same as that of the original signal $x$. The updater $\mathcal{U}(\cdot)$ in Eq 3 corrects this by replacing $x^e$ with smoothed values $s$. Specifically, $\mathcal{U}(\cdot)$ restores the correct running average and thus reduces aliasing:

$$s_k = x_{2k} + (d_{k-1} + d_k)/4 \qquad (14)$$

As shown in Figure 8, $d_k$ is the difference between the odd sample $x_{2k+1}$ and the average of two even samples. This causes a loss $d_k/2$ in the area with the red shade. To preserve the running average, this area is redistributed to the two neighbouring even samples $x_{2k}$ and $x_{2k+2}$, which shapes a coarser piecewise linear signal $s$ in the solid line. The signal after LiftDownPool, drawn as solid line, covers the same area with the original signal dashed line. LiftDownPool reduces aliasing compared to the downsizing drawn in the solid line in (a).

**Out-of-distribution Robustness** We show the robustness of pooling methods for each corruption and perturbation type in Figure 9. Corruption Error (*CE*) is the metric of the robustness to corruptions on ImageNet-C. And Flip Rate (*FR*) is reported for the robustness to perturbation on ImageNet-P. Following (Hendrycks & Dietterich, 2019), we report both unnormalized raw values

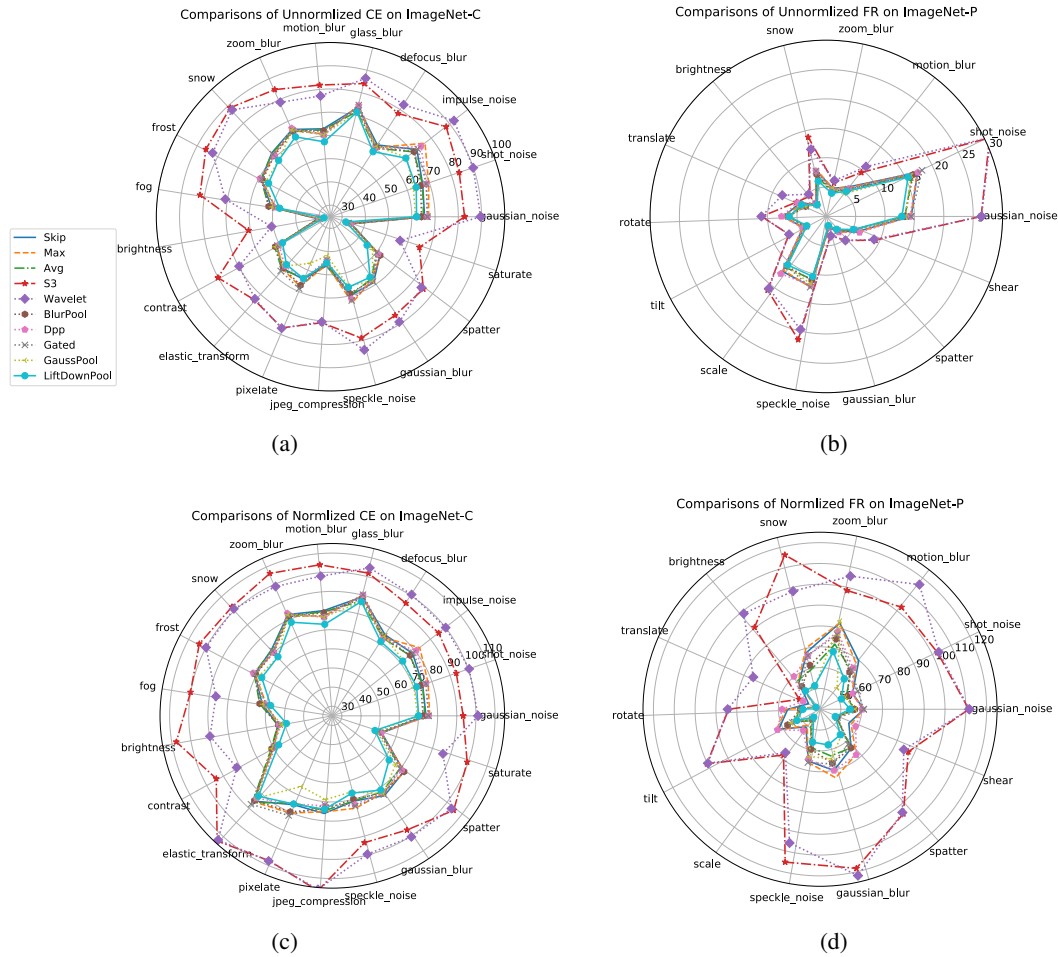

Figure 9: **Comparisons between the robustness of various pooling methods to per kind of corruption on ImageNet-C and perturbation on ImageNet-P.** LiftDownPool presents stronger robustness to almost all the corruptions and perturbations.

and normalized values by AlexNet's *CE* and *FR*. Lower values are better. As seen in Figure 9(a) and (c), LiftDownPool gets the lowest *CE* for most of the "high frequency" corruptions including gaussian noise and spatter, as well as the "low frequency" corruptions such as motion blur, zoom blur. In Figure 9(b) and (d), it clearly shows LiftDownPool has less sensitivity to most of the perturbations such as speckle noise and gaussian blur.

**Visualization** of Up-pooling In Figure 10, we show the feature map for each predicted category from the last layer of SegNet using varying up-pooling methods. Using MaxUpPool, the feature maps look noisy and less continuous due to the fact that MaxUpPool generates the output with many 'zeros', where there is no information. By applying a BlurPool following the MaxUpPool, the feature maps turn more smooth, while still with less details. LiftUpPool, benefiting from the recorded details during LiftDownPool, produces finer feature maps for each category. It has smooth edges, continuous segmentation maps and less aliasing.

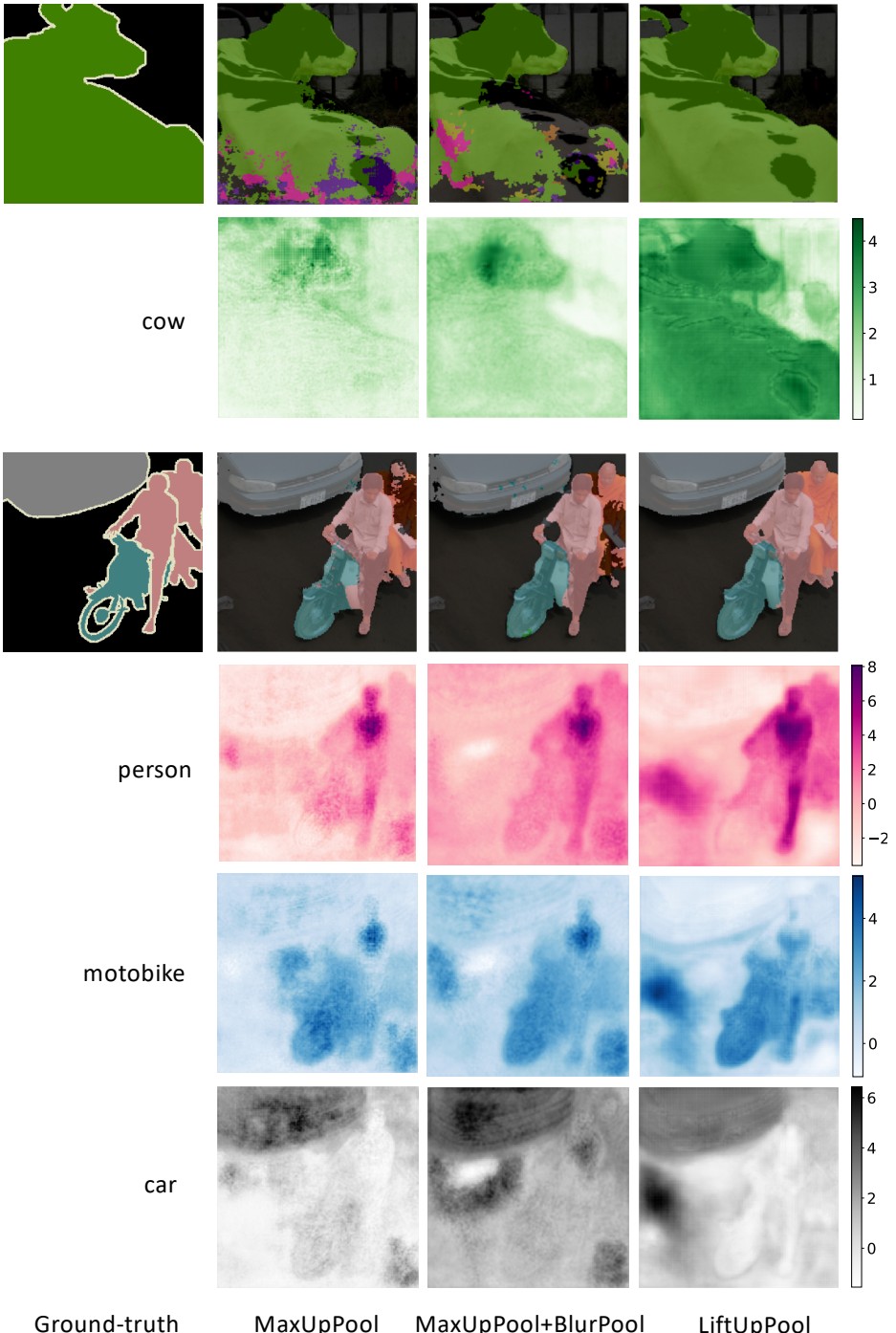

Figure 10: **Visualization of feature maps per-predicted-category from the last layer of SegNet.** Lift-UpPool generates more precise predictions for each category.

