# OpenReview forum: "LiftPool: Bidirectional ConvNet Pooling"
_ICLR.cc/2021/Conference — ICLR 2021 Poster_

### Official Review · AnonReviewer2 · 2020-10-27
**Simple yet effective method, but need more analysis / evidence**

**Rating:** 7
**Confidence:** 4

**Review:**

This paper proposes a simple downscaling / upscaling method LiftPool, inspired by Lifting Scheme from signal processing. Compared with traditional max / avg pooling used in CNNs, LiftPool decomposes input signal into a downscaled approximation and difference component, which can be used for classification (by summing both components) or segmentation (by skip-layer connections). Experiments on both tasks show proposed method yields better performance, and additional analysis on stability indicates LiftPool is more robust than other pooling methods.

Pros:
- A simple but effective method by transferring knowledge from signal processing to vision/ML.
- Pooling is a fundamental block in CNN architecture and this work would benefit a wide range of research and applications
- Consistently better experiment results compared with other commonly used pooling methods, by a significant margin
- Paper is relatively clear written and easy to follow

Cons:
- My main concern with this paper is lacking measurements over # of params, FLOPs and latency. Unlike traditional pooling methods which are usually parameter-free and (relatively) fast to compute, LiftPool does need (from authors) "convolution operators followed by non-linear relu operators" to simulate the filters in  $\mathcal{P}$ and $\mathcal{U}$. The implementation details of these conv operators are missing (except the kernel size is 5, from ablation study). It is unclear whether the performance boost of proposed method is from the effectiveness of LiftPool, or from added capacity of network with more parameters and computations.


Minor comments:
- Abstract: "upsampling a down-scaled feature map loses much information": this is not necessarily true.  Downscaling could lose information while upscaling alone usually don't bring more information.  Moreover, CNNs could still "memorize" spatial information in its depth channel (consider a space-to-depth that reduces spatial resolution but still preserves all information).

- Both $\mathcal{P}$ and $\mathcal{U}$ should be real-valued and using conv + relu might limit filter responses to non-negative values? If multiple conv layers are used and the last one do not have any activation, please specify.

- Eq 4/5 proposes additional loss to help training LiftPool but its effect is not backed up by experiments.

- Figure 3: it might be beneficial to show the original high res feature map together with baseline pooling results (e.g. max / avg pooled) to demonstrate information preserved by LiftPool.

- Experiments conducted are more towards lightweight backbones (ResNet18/50 and MobilenetV2). Would LiftPool also work well with larger architectures? This is specially important given the extra params and computes LiftPool brings.

- On segmentation experiments: It is well-known that bringing low level features in decoders of modern segmentation networks could boost model performance. How would LiftPool work with architectures that have decoder with skip-connections? The results on DeeplabV3plus looks good, but it is more from transferability (pretrained on image classification using liftpool).

- On deeplabv3plus setup: it would be great to clarify more details on this model, e.g. decoder and output stride used in training and validation.

Final review:

The authors addressed most of my concerns. Just a nit that it could be helpful to add the total FLOPs into the table (together with # of params) just for completeness.

---

> ### Author Response · Authors · 2020-11-24
> **Response to AnonReviewer2 (Part2/2)**
>
> **Abstract: "upsampling a down-scaled feature map loses much information": this is not necessarily true.**
>
> Sorry for the unclear statement here. We will modify the sentence to "upsampling a downscaled feature map can not recover the lost information in the downsampling."
>
> **Both $\mathcal{P}(\cdot)$ and $\mathcal{U}(\cdot)$  should be real-valued and using conv + relu might limit filter responses to non-negative values? If multiple conv layers are used and the last one do not have any activation, please specify.**
>
> Yes, the reviewer is correct. $\mathcal{P}(\cdot)$ and $\mathcal{U}(\cdot)$ should be real-valued and we use a 1D convolutional layer with kernel size $K$ followed by a ReLU layer and then another 1D convolutional layer with kernel size 1. We add the new equations Eq. (4) and Eq. (5) in Section 2.1 on Page 3. Please also see the response to comment 1 from AnonReviewer1.
>
> **Eq 4/5 proposes additional loss to help training LiftPool but its effect is not backed up by experiments.**
>
> We add an ablation study on CIFAR100 using VGG13 to illustrate the effect of the additional loss. With the additional losses, LiftPool achieves a lower error rate on CIFAR100 shown in the below table. We add this result in Table 1 in the modified manuscript.
>
> |                         | Top-1 error |   |   |   |
> |-------------------------|-------------|---|---|---|
> | without addition losses | 26.43       |   |   |   |
> | with additional losses  | 24.35       |   |   |   |
>
> **Figure 3: it might be beneficial to show the original high res feature map together with baseline pooling results (e.g. max / avg pooled) to demonstrate information preserved by LiftPool.**
>
> Thanks for the suggestion. Using Resnet50 with input size $224\times 224$, we get the feature maps $112\times 112$ from the first pooling layer. We show the feature maps in the supplementary. Although they are not very high resolution, we see LiftDownPool maintains local structure better than MaxPool.
>
> **Experiments conducted are more towards lightweight backbones (ResNet18/50 and MobilenetV2). Would LiftPool also work well with larger architectures? This is specially important given the extra params and computes LiftPool brings.**
>
> Running experiments on the large-scale ImageNet is heavy for us. For example, using a larger Resnet101 backbone takes at least one week for running 100 epochs with 4 NVIDIA GTX 1080 GPU cards. We are doing the experiments and expect we can add all the results in our final version.
>
> Note our LiftPool does not bring many parameters (only 3K using ResNet50) compared to MaxPool. See also comment 1 above and our response to a similar comment by AnonReviewer1.
>
> **On segmentation experiments: It is well-known that bringing low level features in decoders of modern segmentation networks could boost model performance. How would LiftPool work with architectures that have decoder with skip-connections? The results on DeeplabV3plus looks good, but it is more from transferability (pretrained on image classification using liftpool)**
>
> The proposed LiftPool is simply a replacement for the local pooling layers (down-pool or up-pool). It is independent of a network having skip-connections or not. For example, SegNet doesn’t have skip-connections while DeepLabV3Plus has. Our experiments (Table 7 and 8) show Liftpool works with both architectures.
>
> We apologize for the confusion caused by the subtitle “Transferability” in Table 8 in the original version. Using ImageNet pretrained weights as an initialization is a standard training protocol for segmentation networks. For a fair apple-to-apple comparison, all models reported in Table 8 are transferred from ImageNet pretraining. We change the subtitle “Transferability” to “Semantic segmentation with DeepLabV3Plus” in the modified manuscript.
>
> **On deeplabv3plus setup: it would be great to clarify more details on this model, e.g. decoder and output stride used in training and validation.**
>
> The network architecture we used is same as the one proposed by the original paper (Chen et al. 2018). We clarify in the paragraph Semantic Segmentation with DeepLabV3Plus of Section 4.4. “We leverage the state-of-the-art DeeplabV3Plus-ResNet50 (Chen et al. 2018). The input image has size $512\times 512$. The output feature map of the encoder is $32\times 32$. The decoder upsamples the feature map to $128\times 128$ and concatenates them with the low-level feature map for the final pixel-level classification. Same as before, all local pooling layers are replaced by LiftDownPool.”

---

> ### Author Response · Authors · 2020-11-24
> **Response to AnonReviewer2 (Part1/2)**
>
> We thank AnonReviewer2 for the constructive feedback and sharp comments.
>
> **My main concern with this paper is lacking measurements over # of params, FLOPs and latency. Unlike traditional pooling methods which are usually parameter-free and (relatively) fast to compute, LiftPool does need (from authors) "convolution operators followed by non-linear ReLU operators" to simulate the filters in $\mathcal{P}(\cdot)$ and $\mathcal{U}(\cdot)$. The implementation details of these conv operators are missing (except the kernel size is 5, from ablation study). It is unclear whether the performance boost of proposed method is from the effectiveness of LiftPool, or from added capacity of network with more parameters and computations.**
>
> The reviewer is right, the parameter efficiency is an important benefit of our method. For $\mathcal{P}(\cdot)$ and $\mathcal{U}(\cdot)$ in a 1D LiftPool layer,  we use a 1D convolution layer with kernel size $K$ followed by a ReLU layer and then another 1D convolution layer with kernel size 1. We add the new equations Eq. (4) and Eq. (5) in Section 2.1 on Page 3. Please also see the response to comment 1 and 3 from AnonReviewer1.
>
> We add a new table Table 6 and a paragraph in Section 4.2 for analyzing the parameter efficiency. “ We compare our LiftDownPool to two other parameterized pooling methods using ResNet50 on ImageNet: GFGP and GaussPool. We achieve a lower error rate compared to GFGP and GaussPool with less parameters. With only 3K more parameters than MaxPool, we reduce the error rate by 1.9%. Our performance boost is due to the LiftPool scheme, not the added capacity.” Thank you.
>
> |                                                      | Top-1 err | #Params (M) |   |   |
> |------------------------------------------------------|----------------|----------------|---|---|
> | MaxPool-ResNet50                                     | 24.26          | 25.55          |   |   |
> | GaussPool (Kobayashi, 2019a) w/ author-provided code | 22.95          | 26.14          |   |   |
> | GFGP(Kobayashi, 2019b) w/ author-provided code       | 22.76          | 31.08          |   |   |
> | Ours: LiftDownPool-ResNet50                          | 22.36          | 25.58          |   |   |

---

### Official Review · AnonReviewer4 · 2020-10-27
**Review of "LiftPool: Bidirectional ConvNet Pooling"**

**Rating:** 8
**Confidence:** 4

**Review:**

This paper presents a new pooling layer that is based on the Lifting Scheme from signal processing. It motivates this approach with the desire for reversible pooling functions for certain tasks. The benefits of this reversibility are demonstrated on a semantic segmentation task. As a drop-in replacement for the pooling lawyer in various neural-network backbones, it also outperforms many other pooling layers on classification tasks (ImageNet).

I thought this paper had great collection of analyses: flexibility (choice of pooling band), effectiveness across kernal sizes, generalizability across various backbones, and robustness to corruptions and perturbations.

*I recommend to accept*. While this may be just another pooling layer, it seems a quite well motivated pooling layer coming from a particular need for reversibility.

Some highlights:
- very clear writing
- very well situated in historical and contemporary literature
- exactly the experiments that I would want to see
- the observation that the sub-band that represents vertical details constributes more to classification accuracy than other sub-bands; curious to know whether this holds outside of VGG13 on CIFAR-100.

Some points for improvement:
- the presentation of the lifting scheme and the use of the LL/HL/HH/HH notation is perhaps a little non-intuitive for anyone without previous exposure; you could make it clearer that not all bands would be necessarily used during pooling, but that the information would be retained for reversing the operation.
- on p.7 you say "sift-invariance"; I think you mean "shift-invariance"
- Instead of saying how you believe your findings will stimulate people to think about problems ("These findings may stimulate researchers to rethink", "We believe such findings will stimulate one to think"), it would be better to perhaps make a claim that needs to be evaluated in the future, or to point out exactly what is left unknown or surprising by your findings.
- Figure 7 is not very clear. Think about people with colour blindness. Also, consider two separate graphs side-by-side or one above the other. It isn't clear what the shaded red area is meant to indicate and how it relates to what is "redistributed." I see what you're trying to show, and I actually just think it is a quite difficult thing to visualize, so maybe Figure 7 is the best you can get to, but I would brainstorm some more on this.
- p 12. "Visualiztion"
- p 12. the closing quotation marks around "high frequency" and "low frequency" go the wrong direction
- Figure 9 caption: the hyphenation in LiftUpPool should be customized; it breaks the word at a weird spot
- Figure 8 consider using various line types instead of colors in order to better accomodate people with color blindness
- Throughout: inconsistent hyphenation "downsizing" vs "down-sizing", "up-sampling" vs "upsampling"
- Bibliography: I think you need to force capitalization in some of the titles. See e.g. "pytorch" and "Mobilenetv2"

----------------

Question: When you "combine all the sub-bands by summing them up", do you literally just add up the values from the corresponding indices across each sub-band so that you're still reducing the dimensionality? I am a little surprised that this doesn't *reduce* performance. Can you say more about this? Why does this work?

Question: Why did you only compare against three baseline pooling methods for the corruptions and perturbations instead of the full gamut as in Table 4?

Question: do you have error bars for your experiments? Or did you run them only once each? For which results did you run your own experiments vs reporting numbers from previous literature (particularily in Table 4)?

----------------
My main uncertainty (why I am not giving this a 5 for confidence) is that I cannot be sure this hasn't been proposed in the past, but it is hard to prove a negative. I can say is that this does appear novel to me.

I am also uncertain about the evaluation on the semantic segmentation task. I am familiar with this problem and the evaluations seem reasonable, but I cannot be sure whether the choices of comparator methods are the strongest alternatives.

---

> ### Author Response · Authors · 2020-11-24
> **Response to AnonReviewer4**
>
> We thank AnonReviewer4 for the detailed suggestions and actionable feedback.
>
> **When you "combine all the sub-bands by summing them up", do you literally just add up the values from the corresponding indices across each sub-band so that you're still reducing the dimensionality? I am a little surprised that this doesn't reduce performance. Can you say more about this? Why does this work?**
>
> Indeed, we just sum up the sub-bands so that we don’t increase the channels as clarified in the paragraph Flexibility of Section 4.2. Empirically, we find it works best.
>
> **Why did you only compare against three baseline pooling methods for the corruptions and perturbations instead of the full gamut as in Table 4?**
>
> We add more comparisons on corruption and perturbations in Table 5. We find SpectralPool, MixedPool and GFGP do not work on ImageNet-C and ImageNet-P, thus we did not report their results in Table 5. We clarify this in the paragraph Out-of-distribution Robustness of Section 4.3 in the modified paper.
>
> **do you have error bars for your experiments? Or did you run them only once each? For which results did you run your own experiments vs reporting numbers from previous literature (particularily in Table 4)?**
>
> For the dataset CIFAR100, we run each experiment three times with different initial random seeds during training and report the averaged error rate with the standard deviation in Table 1, 2 and 3.
>
> In Table 4, we run all the experiments by ourselves for an apple-to-apple comparison. S3Pool, WaveletPool, SpectralPool, GatedPool and MixedPool did not report results on ImageNet using ResNets and MobileNet-V2 in their paper. For BlurPool, DPP, GFGP and GaussPool, we use the code released by the authors. As it requires a long time to run experiments on ImageNet, we did not run it multiple times. We will release our code package, including all reported pooling methods.
>
> **the presentation of the lifting scheme and the use of the LL/HL/HH/HH notation is perhaps a little non-intuitive for anyone without previous exposure; you could make it clearer that not all bands would be necessarily used during pooling, but that the information would be retained for reversing the operation.**
>
> We add the suggested explanation in the second last paragraph of Section 2.1 on Page 3. Also see our response to comment 4 from AnonReviewer1.
>
> **on p.7 you say "sift-invariance"; I think you mean "shift-invariance"**
>
>  Corrected it in the modified manuscript. Thank you.
>
> **Instead of saying how you believe your findings will stimulate people to think about problems ("These findings may stimulate researchers to rethink", "We believe such findings will stimulate one to think"), it would be better to perhaps make a claim that needs to be evaluated in the future, or to point out exactly what is left unknown or surprising by your findings.**
>
> We rephrase this sentence in the modified manuscript on Page 6. “Whether the two spatial dimensions should be treated equally, we leave for our future work.”
>
> **Figure 7 is not very clear. Think about people with colour blindness. Also, consider two separate graphs side-by-side or one above the other. It isn't clear what the shaded red area is meant to indicate and how it relates to what is "redistributed." I see what you're trying to show, and I actually just think it is a quite difficult thing to visualize, so maybe Figure 7 is the best you can get to, but I would brainstorm some more on this.**
>
> Thanks for the suggestion. We modified the figure accordingly.
>
> **p 12. "Visualiztion"**
>
> **p 12. the closing quotation marks around "high frequency" and "low frequency" go the wrong direction**
>
> **Figure 9 caption: the hyphenation in LiftUpPool should be customized; it breaks the word at a weird spot**
>
> **Figure 8 consider using various line types instead of colors in order to better accomodate people with color blindness**
>
> **Throughout: inconsistent hyphenation "downsizing" vs "down-sizing", "up-sampling" vs "upsampling"**
>
> **Bibliography: I think you need to force capitalization in some of the titles. See e.g. "pytorch" and "Mobilenetv2"**
>
> Thank you. All corrected, together with a few more repairs.

---

### Official Review · AnonReviewer3 · 2020-10-28
**review 3017**

**Rating:** 5
**Confidence:** 4

**Review:**

This paper introduces a learnable pooling strategy to preserve details when down-sampling feature maps. The proposed LiftPool method is derived from the classical lifting scheme of signal processing, which contains a LiftDownPool to decompose a feature map into various down-sampled sub-bands in the down-sampling process, and a LiftUpPool to merge these sub-bands in the up-sampling process. The proposed pooling strategy is incorporated into existing methods to achieve  better results on image classification and semantic segmentation tasks.


I have the following main concerns.

1. The motivation of this work is not well-explained. First, it is not explained clearly why the pooling operation is desired to be inversible. Second, the main purpose of pooling is to obtain a larger receptive field. While it does lose spatial information, one can seek to other methods to either avoid using pooling (e.g., using dilation convolutions) or enhancing spatial informaiton (e.g., skip connection or feature fusion). Hence, it would be better to highlight the necessity of preserving details and being inversible in pooling operations.

2. Section 2 only describes the proposed method. However, it is important to highlight the novelties in the proposed method. I am not sure whether the method described in section 2 is an easy adaption of the existing Lifting Scheme method. I also do not understand why the proposed LiftDownPool is formulated as the multiplication of f_{split}, f_{predict} and f_{update}.

3. In the second paragraph of section 3, it is said that most existing pooling methods are not inversible. However, this means that there are some methods that are inversible. However, the paper do not explain it.

4. It is not clear which sub-bands should be used as pooled results.

5. The Lift scheme has already been studied in neural networks in (Zheng et al., 2010). The difference is not discussed in depth.

6. The proposed method is only evaluated on the PASCAL-VOC12 (Everingham et al., 2010) with some methods. Evaluations on more datasets and

7. Experiments in Table 6 seems not fair, as the proposed method introduces more learnable parameters.


Minor issues.

1. The title is called "bidirectional". However, after reading the paper, I still do not understand how the propsoed method relates to it.

2. In figure 5, I am not sure how we can evaluate the "Invertibility" by comparing the segmentation results.

---

> ### Comment · AnonReviewer4 · 2020-11-10
> **Regarding motivation and invertibility**
>
> I understood the main motivation for invertible pooling layers to be tasks like segmentation that require pixel-precise classification results. This also explains why they chose comparison of segmentation results as a demonstration of how this invertibility is useful.

---

> ### Author Response · Authors · 2020-11-24
> **Response to AnonReviewer3 (Part 1/2)**
>
> We thank AnonReviewer3 for the precious feedback and suggestions.
>
> **First, it is not explained clearly why the pooling operation is desired to be inversible.**
>
> We believe the invertibility of pooling functions is useful for upsampling feature maps in encoder-decoder architectures for image-to-image translation tasks, like semantic segmentation, super-resolution, and colorization. In existing pooling methods, the lost information during downpooling can not be recovered when uppooling the feature maps. One can try to fix this by using skip-connection or feature fusion. We prefer to address it in the pooling layer itself. We utilize the invertibility of the Lifting Scheme to develop LiftUpPool, which fuses the preserved detailed sub-bands and low-resolution feature maps to generate high-resolution feature maps. Our experiments confirm its benefit for the pixel-wise classification task.
>
> **It would be better to highlight the necessity of preserving details**
>
> Existing pooling methods are lossy. At the same time, it is crucial for pooling layers to keep the activations which are important for the network’s discriminability (Saeedan et al., 2018; Boureau et al., 2010). Dilated convolutions may avoid losing spatial information, but it requires much more parameters than a pooling layer. We prefer to improve the pooling layer itself.
>
> **It is important to highlight the novelties in the proposed method. I also do not understand why the proposed LiftDownPool is formulated as the multiplication of $f_{\text{split}}$, $f_{\text{predict}} $ and $f_{\text{update}}$**
>
> We thank the reviewer for the suggestion. We highlight our novelty of LiftPool in Section 2.1 on Page 3 in the modified manuscript. The classic Lifting Scheme applies pre-defined low-pass filters and high-pass filters to decompose an image into four sub-bands. We implement the Lifting Scheme in our pooling layer by learnable convolutional kernels and make it end-to-end trainable with a deep convolutional network.
>
> We express our LiftDownPool function as $f_{\text{update}}\circ f_{\text{predict}}\circ f_{\text{split}}(\cdot)$, where $\circ$  means the function composition operator (not multiplication). Namely, LiftDownPool consists of three consecutive steps, split a signal into two disjoint sets; predict the detail sub-band; and update the approximation sub-band.
>
> **However, this means that there are some methods that are inversible. However, the paper do not explain it.**
>
> To the best of our knowledge the functions of existing pooling methods are not invertible. We rephrase the sentence accordingly in the second paragraph of Section 3 on Page 5. Thank you.
>
> **It is not clear which sub-bands should be used as pooled results.**
>
> We recommend to sum up all the sub-bands as we observed this gives the best results. We clarify this in the paragraph Flexibility of Section 4.2.
>
> **The Lift scheme has already been studied in neural networks (Zheng et al., 2010). The difference is not discussed in depth.**
>
> We add the discussion in Section 3 Related Work on Page 5.
> “(Zheng et al., 2010) introduce back-propagation for Lifting Scheme to perform nonlinear wavelet decomposition. They propose an update-first Lifting Scheme and use back-propagation to replace the Updater and Predictor in the Lifting Scheme. In this way, they realize a back-propagation neural network in lifting steps for signal processing. There is no pooling layer used. We develop down-pooling and up-pooling layers by leveraging the idea of the Lifting Scheme for image processing. We utilize convolution layers and ReLU layers to implement the Updater and Predictor, which are optimized end-to-end with the deep neural networks. Our developed pooling layers are easily plugged into various backbones.”
>
> **The proposed method is only evaluated on the PASCAL-VOC12 (Everingham et al., 2010) with some methods. Evaluations on more datasets and**
>
> We do not understand this question. We use PASCAL-VOC12 as it is a standard and widely used benchmark for semantic segmentation. We also evaluate pooling methods in the context of image classification on CIFAR100 and ImageNet. And we evaluate the robustness to corruptions and perturbations on ImageNet-C and ImageNet-P.
>
> **Experiments in Table 6 seems not fair, as the proposed method introduces more learnable parameters.**
>
> The parameter-efficiency is indeed an important benefit of our method. We add the suggested parameter comparisons in the updated Table 7. LiftUpPool with only 5K (0.17%) more parameters than MaxUpPool and MaxUpPool+BlurPool, achieves a 6.2% and 4.9% higher mIoU. Thank you.
>
> |                    | mIoU | #Params |   |   |
> |--------------------|------|------------|---|---|
> | MaxUpPool          | 62.7 | 29.45M     |   |   |
> | MaxUpPool+BlurPool | 64.0 | 29.45M     |   |   |
> | LiftUpPool         | 68.9 | 29.50M     |   |   |

---

> > ### Author Response · Authors · 2020-11-24
> > **Response to AnonReviewer3 (Part 2/2)**
> >
> > **The title is called "bidirectional". However, after reading the paper, I still do not understand how the propsoed method relates to it.**
> >
> > We clarify in the Introduction of the modified manuscript: “Different from existing pooling operations, we propose in this paper a bidirectional pooling called LiftPool, including LiftDownPool which preserves correlated structures in feature maps as well as details when downsizing the feature maps, and LiftUpPool for generating finer upsampled feature maps.”
> >
> > **In figure 5, I am not sure how we can evaluate the "Invertibility" by comparing the segmentation results.**
> >
> > We apologize for the confusion of the subtitle “Invertibilty”. We change it to “LiftUpPool for semantic segmentation”.

---

### Official Review · AnonReviewer1 · 2020-10-29
**Good paper about invertible pooling**

**Rating:** 7
**Confidence:** 5

**Review:**

This paper proposes a pooling method for CNNs using a Lifting scheme. The proposed LiftPool consists of LiftDownPool and LiftUpPool. The LiftDownPool decomposes a feature map into four sub-bands (LL/LH/HL/HH). The ListUpPool is reversible to the original features maps from the decomposed components. The decomposition is realized by convolutional layers and relu operation and trained with CNNs. Experimental results show LiftPool achieves better results on image classification and segmentation.
Overall, the paper is well written and easy to follow. The method is well motivated and technically sound. Experimental results are compelling.

Pros
+ LiftPool is well motivated by the difficulty of conducting upsampling (Badrinaryanan et al. 2017) and the aliasing problem (Zhang 2019) of max pooling.
+ LiftPool shows better performance than state-of-the-arts pooling methods in image classification evaluated on the ImageNet dataset with three backbone networks.
+ The robustness of the LiftPool is widely evaluated in terms of out-of-distribution and shift-robustness.
+ The invertible property of LiftPool seems to be very suitable to Encoder-Decoder architecture in semantic segmentation. It shows better scores than MaxUpPool + BlurPool(Zhang, 2019), and the semantic segmentation results are more smooth.

Cons
- Details of P() and U() are not clear. According to p.3, they are convolutional operators followed by non-linear relu operators. How many convolutional operators are used?
- To learn P() and U(), the loss functions of Eq.(4) and Eq.(5) are used. Is it necessary to introduce regularization parameters to balance with the original loss function?
- LiftPool is applied to each of the local pooling of each layer of CNNs. Are different filters learned for each position and layers? How many parameters (trainable and hyper-parameters) are added by LiftPool?
- How to perform 1D-operations along the two dimensions of the image to obtain LL,LH,HL,HL should be clarified, eg., orders.
- Table1 shows that larger kernel sizes perform better. Why did the authors not test larger kernel sizes than 5?
- There is a work that uses Lifting-wavelet for CNNs. The difference between LiftPool to this work should be discussed. \
M.X.B.Rodriguez, A.Gruson, L.F.Polania, S.Fujieda, F.P.Oritz, K.Takayama, T.Hachisuka, Deep Adaptive Wavelet Network, In Proc. WACV2020

Minor problem
- In Sec.2.1, a clear explanation of the correspondence of s,d, and L,H would help readers to understand the LiftPool.

---

> ### Comment · AnonReviewer4 · 2020-11-10
> **I have similar questions**
>
> I didn't focus on this in my (positive) review, but agree that additional details about the operators would be really helpful. What does the actual network structure look like in each of these pooling layers? Could the authors count how many additional parameters this introduces? (It doesn't seem much; just a convolution pattern, P, and U. So, it should be roughly O(c * k^2) per pooling layer where k is the kernal size and c is the number of channels? And I think something like 3 * c * k^2? I might be missing something though.)

---

> ### Author Response · Authors · 2020-11-24
> **Response to AnonReviewer1**
>
> We thank AnonReviewer1 for the helpful comments and suggestions.
>
> **Details of $\mathcal{P}(\cdot)$ and $\mathcal{U}(\cdot)$  are not clear.**
>
> For $\mathcal{P}(\cdot)$ and $\mathcal{U}(\cdot)$ in a 1D LiftPool layer, we use a 1D convolution layer with kernel size $K$ followed by a ReLU layer and then another 1D convolution layer with kernel size 1. We add the details in our modified manuscript, including two new equations Eq. (4) and Eq. (5) in Section 2.1 on Page 3.
> “ $\mathcal{P}(\cdot) = \text{Conv(kernel=1,stride=1,groups=$C$)} \circ \text{ReLU()} \circ \text{Conv(kernel=$K$,stride=1,groups=$C$)}$
>  $\mathcal{U}(\cdot) = \text{Conv(kernel=1,stride=1,groups=$C$)} \circ \text{ReLU()} \circ \text{Conv(kernel=$K$,stride=1,groups=$C$)}$
>
> Where $K$ is the kernel size and $C$ is the number of input channels. ”
> For a 2D LiftPool, it contains a 1D LiftPool operation performed in the horizontal direction and two 1D LiftPool operations performed in the vertical direction.
>
> **Is it necessary to introduce regularization parameters to balance with the original loss function?**
>
> Indeed, we introduce regularization parameters $\lambda_u$ and $\lambda_p$ to balance the additional losses and the original loss in both the image classification task and semantic segmentation task. We clarify in the modified manuscript: Eq. (8) in Section 2.1 on Page 3 and Experiment Settings in the supplementary.
>
> **Are different filters learned for each position and layers? How many parameters (trainable and hyper-parameters) are added by LiftPool?**
>
> We clarify in the paragraph Parameters Efficiency of Section 4.2: “For all pooling layers in one network, we use the same kernel size in LiftPool. Besides the two hyper-parameters: the kernel size $K $and the balancing weights $\lambda_u$ and $\lambda_p$ for the losses, there are no extra hyper-parameters introduced. For the trainable parameters, recall $\mathcal{P}(\cdot)$ and $\mathcal{U}(\cdot)$ has 1D convolutions, so each has $C\times K+C$ parameters where $K$ is the kernel size of the first 1D convolution and $C $ is the number of the input channels. A 2D LiftPool has $3(C\times K+C)$ parameters. ”
>
> **How to perform 1D-operations along the two dimensions of the image to obtain $LL$,$LH$,$HL$,$HL$ should be clarified, eg., orders**
>
> We clarify in the second last paragraph of Section 2.1 on Page 3: “Following the standard Lifting Scheme, we first perform a LiftDownPool-1D along the horizontal direction to obtain an approximation part $s$ (low frequency in the horizontal direction) and a difference part $d$ (high frequency in the horizontal direction). Then, for each of the two parts, we apply a LiftDownPool-1D along the vertical direction. By doing so, $s$ is further decomposed into $LL$ (low frequency in vertical and horizontal directions) and $LH$ (low frequency in the vertical direction and high frequency in the horizontal direction). $d$ is further decomposed into $HL$ (high frequency in the vertical direction and low frequency in the horizontal direction) and $HH$ (high frequency in vertical and horizontal directions). We can flexibly choose sub-band(s) for down-pooling and keep the other sub-band(s) for reversing the operation.” Thank you.
>
> **Table1 shows that larger kernel sizes perform better. Why did the authors not test larger kernel sizes than 5?**
>
> We did more experiments using VGG13 on CIFAR100 in Table 2 on Page 6. The top-1 error is 24.40% with kernel size 7 and 24.35% with kernel size 5. A kernel size larger than 5 does not bring performance gain and introduces more computations, thus we prefer kernel size 5, and indicated by our motivation in the updated paragraph on Effectiveness in Section 4.2 on Page 7.
>
> **There is a work that uses Lifting-wavelet for CNNs. The difference between LiftPool to this work should be discussed.
> M.X.B.Rodriguez, A.Gruson, L.F.Polania, S.Fujieda, F.P.Oritz, K.Takayama, T.Hachisuka, Deep Adaptive Wavelet Network, In Proc. WACV2020**
>
> We clarify in Section 3 Related work: “Rodriguez et al. (2020) introduce the Lifting Scheme for multiresolution analysis in a network. Specifically, they develop an adaptive wavelet network by stacking several convolution layers and Lifting Scheme layers. They focus on an interpretable network by integrating multiresolution analysis, rather than pooling. Our paper aims at developing a pooling layer by utilizing the lifting steps. We develop a down-pooling which constructs various sub-bands with different information, and an up-pooling which generates refined upsampled feature maps.”
>
> **In Sec.2.1, a clear explanation of the correspondence of $s$,$d$, and $L$,$H$ would help readers to understand the LiftPool.**
>
> Thanks for the suggestion. We provide more explanation on $s$,$d$ and $LL$, $LH$, $HL$, $HH$ in Section 2.1 on Page 3. See also our response to similar comment above.

---

### Decision · Program_Chairs · 2021-01-07
**Final Decision**

**Decision:**

Accept (Poster)

**Comment:**

The paper presents a bidirectional pooling layer inspired by the classical Lifting scheme from signal processing. LiftDownPool is able to preserve structure and details in different sub-bands, whereas LiftUpPool is able to generate a refined up sampled feature map using the detail sub-bands. This is very useful for image to image translation tasks and all tasks that involve up scaling.
This is a solid contribution with extensive and thorough experiments and direct practical usage, clear accept.